# Spatiotemporal Dynamics of Ecological Total-Factor Energy Efficiency and Their Drivers in China at the Prefecture Level

**DOI:** 10.3390/ijerph16183480

**Published:** 2019-09-18

**Authors:** Guangdong Li

**Affiliations:** 1Key Laboratory of Regional Sustainable Development Modeling, Institute of Geographic Sciences and Natural Resources Research (IGSNRR), Chinese Academy of Sciences (CAS), 11A Datun Road, Chaoyang District, Beijing 100101, China; ligd@igsnrr.ac.cn; 2College of Resources and Environment, University of Chinese Academy of Sciences, Beijing 100049, China

**Keywords:** ecological total-factor energy efficiency, ecological total-factor energy productivity, influencing factor, epsilon-based measure, spatial panel data model, China

## Abstract

Improvement of ecological total-factor energy efficiency (ETFEE) is crucial for transformation of China’s economic growth pattern, energy conservation and emissions abatement. Here we combined the epsilon-based measure (EBM) and the Global Malmquist-Luenberger (GML) productivity index to evaluate ETFEE and ecological total-factor energy productivity (ETFEP) and its decompositions for 283 prefecture-level cities in China between 2003 and 2013. A spatial econometric model is used to investigate factors influencing ETFEE and ETFEP. Results indicated that ETFEE, ETFEP and corresponding trends differ significantly depending on whether environmental constraints are considered. No convergence trend was found in ETFEE between prefecture-level cities. Technical progress plays the largest role in increasing ETFEP growth. Pure efficiency change and scale efficiency change, however, are the main hindering factors. Boosting cumulative technological progress, cumulative scale efficiency growth rate and cumulative pure efficiency growth rate are important means of increasing ETFEP. I also found that areas with high levels of economic development do not completely overlap with areas of high ETFEE. Surprisingly, the fiscal expenditure on scientific undertakings and technological spillover effects from foreign direct investment (FDI) have not substantially increased ETFEE. Whereas increased industrialization hinders the improvement of ETFEE. Furthermore, reducing per capita energy consumption help boost ETFEE. In addition, endowment advantages of factors of production have a positive overall effect on improving ETFEE. Lastly, important policy implications are inferred.

## 1. Introduction

China already consumes the most energy and emits the most carbon of any country in the world [1,2]. Improving energy efficiency will be key to meeting China’s energy conservation and emissions abatement targets as well as controlling its atmospheric pollution [3]. Economists widely agree that if economic growth is primarily driven by inputs of factors of production, then it is extensive and unsustainable. If, on the other hand, total-factor productivity generates major contributions to economic growth, economic growth will be intensive and sustainable [4]. As such, striving to improve energy efficiency and energy total-factor productivity as well as transforming the economic growth pattern in pursuit of sustainable development is not only key to ensuring that China achieves its target for Intended Nationally Determined Contributions (INDC) and high-quality economic development, but is also a topic of global concern.

Traditional total-factor productivity evaluation only considers essential productive factors such as labor and capital, energy and the resource environment factors were usually ignored, which can distort assessments of social welfare changes and economic performance, resulting in misguided policy recommendations [5]. Therefore, greater interest has been focused in recent years on comprehensive accounting of resource environment losses in total-factor energy efficiency evaluation [6]. Hu and Wang [6], for example, presented a method for calculating a total-factor energy efficiency index via data envelopment analysis (DEA), which includes energy as an input factor in energy efficiency accounting. Specifically, it can be defined as the ratio of the target energy input that is suggested by the DEA to the actual energy inputs in a region. This method then calculates the gap between the target and actual energy inputs. Other inputs were suggested along with energy input for consideration in energy efficiency measurement. It has become a mainstream method in energy efficiency evaluation, but its drawback is that it does not consider undesirable outputs (these are usually related to externalities, side effects, and adverse impacts produced by or resulting from the course of operations, such as emission, pollution, waste, etc.). In response to the growing concern surrounding environmental effects, the ETFEE, which is generally defined as the ratio of the target energy input with undesirable outputs to the actual energy input in a region, namely, incorporating undesirable outputs into energy efficiency analysis, has been proposed in the past few years [7,8,9]. Li and Hu [10], for example, used a slack-based model (SBM) to calculate the ETFEP of 30 provinces in China between 2005 and 2009, taking into account undesirable outputs. Zhang and Choi [8] also used an SBM to compute ecological energy efficiency between 2001 and 2010 in China, taking into account the undesirable outputs of CO_2_, SO_2_ and chemical oxygen demand [11]. Lin and Tan [7], meanwhile, used a meta-frontier slack-based DEA model to calculate the ETFEP and energy saving potential for four of most energy-intensive industries in China. Li and Lin [12] went further by calculating and comparing the ETFEP of heavy and light industries and assessing their technological gap. Yang and Zhang [13] used the bootstrapping method and an SBM to analyze the ecological energy efficiency of China for 30 provinces from the perspectives of resources, the environment and economic performance. A review of existing literature shows that slack-based models with undesirable outputs are considered, have been commonly used in studies on ETFEE evaluation in China at the provincial level.

The keys to calculating ETFEE are dealing with undesirable outputs and choosing the right method. There are currently four methods for quantifying the influence of environmental pollution emissions (or undesirable outputs) on energy efficiency performance [14,15,16,17]. The first is to treat undesirable outputs is to simply disregard them from the production function. The second is to treat environmental pollution as a direct input factor [18]. This method, however, does not reflect real production processes. The third method is to treat undesirable outputs as normal outputs of the production function. Because this method is more consistent with actual production processes, it has been employed extensively in recent years. The fourth is to transform the undesirable outputs (such as monotone decreasing transformation, data normalization, and linear transformation). With respect to ecological energy efficiency evaluation, radial and non-radial DEA models are the most common approaches, though both have their drawbacks. Radial models ignore non-radial slack, which leads to biased measurements, and these deal mainly with proportional changes of input or output variables, which is difficult to achieve in practice [19]. On the contrary, non-radial models can obtain non-radial slack, which is missed by radial models, but they result in a loss of proportionality in the original data. It is, therefore, necessary to combine radial and non-radial models to calculate efficiency more rationally. In order to achieve this, Tone and Tsutsui [19] set forth an EBM model that integrates radial and non-radial features under a uniform framework. The EBM model effectively addresses the defects of radial and non-radial models and has been effectively applied to calculate various types of ETFEE in the recent past. Qin et al. [20] used the EBM model and the GML productivity index, for example, to calculate changes in the stationary energy efficiency and actional energy efficiency of coastal provinces in China between 2000 and 2012. Yang, Wang and Geng [9] utilized the EBM model to estimate the ETFEE of China for 30 provinces, as well as the eastern, western and central regions between 2007 and 2015. Given the advantages of the EBM model, this study combines it with the GML productivity index to calculate corresponding ETFEE, ETFEP and its decompositions.

However, previous studies have several limitations worth noting: (1) It should be noted that due to missing data at the prefecture level, most current analyses are investigated at the provincial level, with only a few prefecture-level cities of China having been investigated [21,22]. It is difficult to obtain information on dynamic changes in ETFEE and ETFEP at the prefecture level for the whole country. (2) The role of environmental constraints in total-factor energy efficiency has not been examined. The specific impact mechanism of undesirable outputs on total-factor energy efficiency has not been fully understood. (3) Spatial spillover or the spatial effect of ETFEE and ETFEP have not gained the deserved attention from the existing literature [23,24], resulting in a biased estimation result. (4) The traditional Malmquist-Luenberger productivity index can only offer relative ETFEP scores, but fails to give specific factors for the changes. In addition, the global dynamic variations of ETFEP has not been measured in the previous literature.

To cope with the above limitations, the contributions and novelties of this research focused on these four aspects. (1) This study realized a downscaling analysis from traditional province-level to prefecture-level. Although some studies have observed the ETFEE of China’s national, regional and provincial levels, there is insufficient research on prefecture-level cities at the national scale [21,22]. The main reason for this is that China’s government only releases energy statistics annually both at the national and provincial level. Only a few prefecture-level cities publish their data, causing a deficiency in the data in the prefecture-level energy statistics. The absence of such data results in the impossibility of achieving some of the investigations at finer spatial scales, particularly at the prefecture level. This then causes the intra-provincial differences of ETFEE to be impossible to reveal. To fix these weaknesses, we utilized a routine remote sensing estimation method to obtain energy use data at the prefecture-level in China by integration of DMSP/OLS (Defense Meteorological Satellite Program/ Operational Line Scanner) ‘city lights’ satellite data and statistical data. We can use energy consumption as a crucial input variable for 283 prefecture-level cities with the aid of this innovative approach. (2) The impact of environmental constraints on the total-factor energy efficiency was deeply analyzed. The traditional research outline of total-factor energy efficiency was empirically extended to ETFEE at the prefecture level by considering undesirable environmental pollution outputs. No studies to date have investigated the effect of environmental constraints on the total-factor energy efficiency at the prefecture level by comparing whether environmental outputs were considered in the estimation of total-factor energy efficiency, thereby allowing an examination of the trends and essential reasons of changes in ecological total-factor energy efficiency. (3) Ubiquitous spatial spillover of ETFEE and ETFEP was investigated and handled properly. Although numerous studies have testified the importance of spatial spillover in neighboring relation of ETFEE, the necessity of addressing spatial effects as well as the detailed influencing mechanism of spatial effects on the ETFEE were not fully understood. In this study, we will contribute to the existing literature by using spatial econometric models to verify and examine the spatial effects of neighboring socio-economic factors of cities on their ETFEE. (4) A novel methodology system was used to dynamically assess and compare the ETFEEs of prefecture-level cities. Following [20,22], a global benchmark technology (GBT) was combined with the epsilon-based measure [25] and GML productivity index, which can obtain interannual comparable efficiency values in comparison with the traditional method, was employed to estimate ETFEE, ETFEP, and its decompositions.

Based on the above analysis, the key research issues or main points were condensed into several aspects, including characterization of evolutionary process and spatial-temporal pattern of ETFEE and ETFEP in China at the prefecture-level cities, identification and estimation of the influencing factors of these two indices changes, investigation of spatial spillover existed in the variation of ETFEE and ETFEP. Specifically, this study selects labor, capital stock and energy use as input factors [9,13,26], constant-price GDP (gross domestic product) as a desirable output [9,13,26], and wastewater discharge, sulfur dioxide emissions and soot (dust) emissions as undesirable outputs [13,22,23]. Then, the EBM and the GML productivity indexes were applied to calculate ETFEE, ETFEP index and constituent components for 283 prefecture-level cities in China between 2003 and 2013, with consideration given to resource and environmental factors. Finally, I utilize a spatial econometric model (the main purpose is to control the spatial effect to obtain a more reliable regression result) to examine the impact of socio-economic factors on the changes in ETFEE and ETFEP.

The rest of this paper structured as follows. In Section 2, a detailed description for the materials and methods is presented. Results analysis and related discussion is illustrated in Section 3. Section 4 concludes this paper and provides several important policy implications.

## 2. Materials and Methods

### 2.1. Calculating Ecological Total-Factor Energy Efficiency and Ecological Total-Factor Energy Productivity

Following [20,22], we constructed a model to calculate ETFEE and ETFEP for cities at the prefecture level in China. A prefectural level city in China is often not a “city” in the usual sense of the term (i.e., a large continuous urban settlement), but instead an administrative unit comprising, typically, a main central urban area (a city in the usual sense, usually with the same name as the prefectural level city), and its much larger surrounding rural area, containing many smaller cities, towns and villages.

#### 2.1.1. Directional Distance Function

Chambers et al. [27] devised the directional distance function (DDF) to evaluate the efficiency of undesirable outputs on account of Luenberger’s benefit function. It is assumed that decision-making unit (DMU) *j* uses input elements *x = (x*_1*j*_*, x*_2*j*_*, …, x_Lj_)*, producing the desirable outputs *y = (y*_1*j*_*, y*_2*j*_*, ..., y_Mj_)* and undesirable outputs *b = (b*_1*j*_*, b*_2*j*_*, …, b_Ij_)*. Following Chung et al. [28], the DDF can be defined as follows:(1)$${\vec{D}}_{0}(x,y,b;g)=\sup\{\beta :(x,y+\beta g_{y},b-\beta g_{b})\in P(x)\}$$
where the non-zero vector *g = (g_y_, g_b_)* is the direction vector of undesirable and desirable outputs. *P(x)* is the set of production possibilities. In the case of a given input *x*, the desirable and undesirable outputs are proportionally increased and decreased, and the maximum multiple of increase in the desirable output *y* and diminution in the undesirable output *b* is *β*. This is used to increase desirable outputs and reduce undesirable outputs.

#### 2.1.2. Global Epsilon-Based Measure Model with Undesirable Outputs

To overcome the inadequacies of radial and non-radial models, Tone and Tsutsui [19] proposed an EBM model that mixes radial and non-radial features in a uniform framework. For the purposes of this study, we assume that there are *N* DMUs, and each DMU uses *L* inputs to produce *Q* outputs. This output-oriented EBM model (the output-oriented EBM model was chosen to accurately determine the weight of different environmental pollutants) can be defined as:(2)$$\begin{array}{l} \min\;\;\gamma^{*}=\frac{1}{\varphi +\varepsilon \sum_{r=1}^{Q}\frac{w_{r}^{+}s_{r}^{+}}{y_{r0}}}\\ s.t.\left\{ \begin{array}{l} \sum_{j=1}^{N}\lambda_{j}x_{ij}\leq x_{i0}\\ \sum_{j=1}^{N}\lambda_{j}y_{ij}-\varphi y_{r0}-s_{r}^{+}=0\\ \lambda \geq 0,\;\;s^{+}\geq 0\\ j=1,2,\ldots ,N;\;\;i=1,2,\ldots ,L;\;\;r=1,2,\ldots ,Q \end{array}\right. \end{array}$$
where $w_{r}^{+}$ is the weight of output *r*, which satisfies $\sum_{r=1}^{Q}w_{r}^{+}=1\;(w_{r}^{+}>0)$; *λ* is the intensity vector; $s_{r}^{+}$ represents the non-radial output relaxation term; subscript “*0*” signifies the evaluated DMU; and *ε* represents the significance of non-radial relaxation relative to the radial *φ*. The key parameter *ε* has a range of [0,1]. When *ε* is 0, the above model is a radial model, and it is a non-radial model when *ε* is 1. To examine the temporal trends of environmental efficiency, global benchmark technology (GBT) is integrated into the EBM model. When taking account of undesirable outputs, the global EBM model can be expressed as:
(3)$$\begin{array}{l} \min\;\;\gamma^{*}=\frac{1}{\varphi +\varepsilon \times \frac{1}{2}\left(\sum_{t=1}^{T}\sum_{m=1}^{M}\frac{w_{m}^{y}s_{my}^{t}}{y_{m0}^{t}}+\sum_{t=1}^{T}\sum_{k=1}^{I}\frac{w_{k}^{b}s_{kb}^{t}}{b_{k0}^{t}}\right)}\\ s.t.\left\{ \begin{array}{l} \sum_{t=1}^{T}\sum_{j=1}^{N}\lambda_{{}_{j}}^{t}x_{{}_{ij}}^{t}\leq x_{{}_{i0}}^{t}\\ \sum_{t=1}^{T}\sum_{j=1}^{N}\lambda_{{}_{j}}^{t}y_{{}_{mj}}^{t}-\phi y_{{}_{m0}}^{t}-s_{my}^{t}=0\\ \sum_{t=1}^{T}\sum_{j=1}^{N}\lambda_{{}_{j}}^{t}b_{{}_{kj}}^{t}-\phi b_{{}_{k0}}^{t}+s_{kb}^{t}=0\\ \lambda \geq 0,\;\;s_{{}^{k}}^{b}\geq 0,\;\;s_{{}^{m}}^{y}\geq 0\\ j=1,2,\ldots ,N;\;\;i=1,2,\ldots ,L;\;\;m=1,2,\ldots ,M;\\ k=1,2,\ldots ,I;\;\;t=1,2,\ldots ,T \end{array}\right. \end{array}$$
where $w_{m}^{y}$ and $w_{k}^{b}$ are the weights of desirable output variables and undesirable output variables, which satisfy both $\sum_{m=1}^{M}w_{m}^{y}=1$ and $\sum_{k=1}^{I}w_{k}^{b}=1$($w_{m}^{y}\geq 0$$w_{k}^{b}\geq 0$); $s_{my}^{t}$ and $s_{kb}^{t}$ correspond to the relaxation of the desirable output *m* and the undesirable output *k*; $x_{{}_{ij}}^{t}$, $y_{{}_{mj}}^{t}$ and $b_{{}_{kj}}^{t}$ represent the *i*th input of the *j*th DMU at time *t*, the *m*th desirable output and the *k*th undesirable output.

#### 2.1.3. Global Malmquist-Luenberger Productivity Index

The GML productivity index is more stable than the traditional Malmquist-Luenberger (ML) productivity index [29]. Generally, a GBT can be defined as $P^{G}=P^{1}\cup P^{2}\cup \ldots \cup P^{T}$. The GBT builds a reference from panel data that includes all the benchmarking techniques for the same period, which embodies its unique advantage [30].

A GML productivity index on account of the GBT overcomes the shortcomings of traditional ML productivity indexes. It is generally defined as follows:(4)$$GML_{t}^{t+1}=\frac{1+{\vec{D}}_{0}^{G}(x^{t},y^{t},b^{t};y^{t},-b^{t})}{1+{\vec{D}}_{0}^{G}(x^{t+1},y^{t+1},b^{t+1};y^{t+1},-b^{t+1})}$$
where $x^{t}$, $y^{t}$ and $b^{t}$ signify the inputs, desirable outputs and undesirable outputs of a prefecture-level city at sample time *t*. ${\vec{D}}_{0}^{G}$ represents the solution to DDF for the GBT set $P^{G}$, and is solved via Model (1). Usually, the GML productivity index can be decompose into two parts: technological change (GTCH) and efficiency change (GECH). GECH can be separated into pure efficiency change (GPEC) and scale efficiency change (GSEC). The specific formula for breaking them down is as follows:
(5)$$\begin{array}{lll} GML_{t}^{t+1} & = & \left[\frac{1+{\vec{D}}_{c}^{G}(x^{t},y^{t},b^{t};y^{t},-b^{t})}{1+{\vec{D}}_{c}^{t}(x^{t},y^{t},b^{t};y^{t},-b^{t})}\times \frac{1+{\vec{D}}_{c}^{t+1}(x^{t+1},y^{t+1},b^{t+1};y^{t+1},-b^{t+1})}{1+{\vec{D}}_{c}^{G}(x^{t+1},y^{t+1},b^{t+1};y^{t+1},-b^{t+1})}\right]\\ & \times & \frac{1+{\vec{D}}_{\upsilon }^{t}(x^{t},y^{t},b^{t};y^{t},-b^{t})}{1+{\vec{D}}_{\upsilon }^{t+1}(x^{t+1},y^{t+1},b^{t+1};y^{t+1},-b^{t+1})}\\ & \times & \left[\frac{1+{\vec{D}}_{c}^{t}(x^{t},y^{t},b^{t};y^{t},-b^{t})}{1+{\vec{D}}_{c}^{t+1}(x^{t+1},y^{t+1},b^{t+1};y^{t+1},-b^{t+1})}\times \frac{1+{\vec{D}}_{\upsilon }^{t+1}(x^{t+1},y^{t+1},b^{t+1};y^{t+1},-b^{t+1})}{1+{\vec{D}}_{\upsilon }^{t}(x^{t},y^{t},b^{t};y^{t},-b^{t})}\right]\\ & = & GTCH_{t}^{t+1}\times GPEC_{t}^{t+1}\times GSEC_{t}^{t+1} \end{array}$$
where ${\vec{D}}_{c}$ and ${\vec{D}}_{\upsilon }$ signify the DDF under constant return-to-scale (CRS) and variable return-to-scale (VRS). When GML is greater than 1, it means that productivity is growing, and when it is less than 1, it means it is falling. When GTCH is greater than 1, equal to 1 or less than 1, it signifies that technological change is increasing, stable or decreasing, respectively. When GPEC (i.e., management level) is greater than 1, it represents an increase in pure efficiency, and when it is less than 1, it represents a decrease. GSEC represents scale efficiency, but it also reflects whether the DMU is maintaining the most suitable level of investment. When GSEC is exceeding, equal to or less than 1, it suggests that its level is increasing, stable or decreasing, respectively.

### 2.2. Analyzing Method of Influencing Factors

#### 2.2.1. Analysis Model of Influencing Factors

Based on the existing literature [10,22,23,24,31], as well as the availability of data on prefecture-level cities in China, this study identified the following influencing factors:

Economic growth (ln*GDPPC*): Following Antweiler*,* et al. [32], we use GDP per capita as a proxy [22] for environmental regulations to test the Porter hypothesis [33].

Factor of production endowment (ln*capitalpc* and ln*energypc*): Capital, labor and energy are the basic factors of production of a prefecture-level city, and they determine its level of productivity. There are significant disparities in endowments of capital, labor and energy across the country, which inevitably affect the production efficiency of cities, which in turn affects ETFEP. This paper uses the capital–labor ratio logarithm (lncappc = ln (K/L)) [31] and per capita energy use (ln*energypc*) to represent endowments of factors of production.

Technological progress, pure efficiency growth and pure scale efficiency growth (ln*GTCH*, ln*GPEC* and ln*GPSC*): Breaking down ETFEP, the three core influencing factors are technological progress, pure efficiency growth, and pure scale efficiency growth. Nevertheless, the aforementioned breakdown of the index only gives the increase from the previous year and does not reflect the growth of each factor during the entire study period. As such, using 2004 as the baseline, this study calculates cumulative technological progress, cumulative pure efficiency growth and cumulative pure scale efficiency growth relative to 2004.

Fiscal expenditure on scientific undertakings (ln*science*): The level of government spending on scientific undertakings usually has an important effect on improvements in science and technology, which can often improve production efficiency. This study considers fiscal expenditure on scientific undertakings as a percentage of GDP. It is expected that the level of expenditure will correlate positively with ETFEP.

Industrial structure (ln*GDP*2): Secondary industry is the largest primary fossil energy consumer. China is in the accelerated industrial development stage, so its industry has a clear internal structural bias, is energy-intensive, and produces high levels of emissions, which directly affect ETFEE [10]. As a result, the added value of secondary industry as a proportion of GDP has been chosen to reflect the industrial structure, with the coefficient expected to be negative.

Fiscal decentralization (ln*decentralization*): China’s current environmental problems can primarily be attributed to its extensive economic development model, which is derived from the government’s “Chinese-style decentralization” approach. Under Chinese-style fiscal decentralization, local governments are responsible for developing their local economies, as well as improving people’s standard of living and protecting the environment. They must, therefore, coordinate economic growth and environmental conservation. It is necessary and vital to examine the impact of Chinese-style fiscal decentralization on energy total-factor productivity. In this study, local fiscal revenue/local fiscal expenditure is used to reflect the degree of fiscal decentralization. It is expected that greater fiscal decentralization will significantly increase ETFEE.

Foreign direct investment (ln*FDI*): FDI is a fundamental factor to consider when looking at China’s ETFEP, but there is currently no consensus on the matter. There are three main hypotheses. First, the “pollution shelter” hypothesis holds that whether heavily polluting industries move to less developed countries as foreign capital depends on whether those countries have lower environmental standards than industrialized countries. Second, the “pollution halo” hypothesis holds that FDI generally has higher technological efficiency than host-country enterprises, making it better able to improve local environmental conditions through technological spillover effects. Third is the comprehensive effect theory, which holds that the environmental effects of FDI are complex and multi-dimensional, as foreign companies can influence the environmental quality of the host country through various means (technology, scale, regulation, structure, etc.). In this study, FDI was defined as a proportion of GDP to quantify the influence of FDI on ETFEP. It is projected that it will have a positive influence coefficient.

The data sample used in this study covers 283 cities at the prefecture level in China between 2003 and 2013. The above control variables are all obtained from the *China Urban Statistical Yearbook* and the *China Regional Economic Statistical Yearbook*.

The above influencing factors can be combined to construct the following regression models:(6)$$\ln ETFEE_{it}=\ln a_{0}+\beta \ln X_{it}+\varphi_{it}+\varepsilon_{it}$$
(7)$$\ln ETFEP_{it}=\ln a_{0}+\beta \ln X_{it}+\varphi_{it}+\varepsilon_{it}$$
where ln ETFEE*_it_* and ln*ETFEP_it_* are the ETFEE and ETFEP of prefecture-level city *i* in year *t* after natural logarithmization, respectively; ln*a*_0_ is a constant; and *β* is the estimated parameter of each influencing factor. ln*X_it_* is the explanatory variables of prefecture-level city *i* in year *t* after natural logarithmization. *φ_it_* is the spatial and period-time fixed effect, used to control the influence of both and reduce missing variable bias. *ε_it_* is the error term.

#### 2.2.2. Model Estimation

Many studies have already found that ETFEE and ETFEP have significant spatial autocorrelation and spatial spillovers [23,24]. Using a traditional OLS model could lead to bias, so we considered it more appropriate to use a spatial econometric model to estimate models. According to Elhorst [34], currently three elementary spatial panel data models can be utilized to address spatial correlation. Below, the ETFEE model is used as an example to demonstrate model estimation, though the process is similar for the ETFEP model. The one concern would be to ensure the explanatory variables data must change to inter-annual change rates to match the ETFEP. In addition, the ETFEP models involve a different time-period, from 2004 to 2013.

The first model is the spatial autoregressive regression (SAR) or spatial lag model (SLM), which assumes that the value of the dependent variable for a particular location depends in part on the value of the dependent variable of its neighbors, which has been weighted with spatial weighting. It is assumed in this study that the level of ETFEE of a prefecture-level city depends in part on the level of ETFEE of its neighboring cities. The SAR can be defined as follows:(8)$$\ln ETFEE_{it}=\delta \sum_{j=1}^{N}w_{ij}\ln ETFEE_{it}+\beta x_{it}+\mu_{i}+\eta_{t}+\varepsilon_{it}$$
where the parameter *δ* is the spatial autoregressive coefficient, which mirrors the degree of impact of spatial factors on the dependent variable. *W_ij_* is the spatial weight matrix of *N × N*. In this study, the Euclidean distance function is used to calculate the spatial weight matrix. Proximity affects neighboring cities’ ETFEE on the ETFEE of the target prefecture-level city. *X_it_* is the independent variable matrix of an *NT × M*, assuming that the number of independent variables is *m*. *μ_i_* is the individual effect of the urban unit. *η_t_* is the time effect. *ε_it_* is the error term and is subordinate to i.i.d. (0, *σ*^2^).

The second model is the spatial error model (SEM). This model is used to verify spatial dependence in the error term and measure the degree of influence of the erroneous dependent variable of neighboring cities on the target prefecture-level city’s dependent variable. The SEM can be expressed as:(9)$$\begin{array}{l} \ln ETFEE_{it}=\beta x_{it}+\mu_{i}+\eta_{t}+\phi_{it}\\ \phi_{it}=\lambda \sum_{j=1}^{N}w_{ij}\phi_{it}+\varepsilon_{it} \end{array}$$
where *ϕ_it_* is the spatial autocorrelation error term, and *λ* reflects the spatial autocorrelation coefficient of the error term.

The third model, the spatial Durbin model (SDM), has a more general application. It includes both the dependent and independent variables of the spatial lag term. It is defined as follows:(10)$$\ln ETFEE_{it}=\rho \sum_{j=1}^{N}w_{ij}\ln ETFEE_{it}+\beta x_{it}+\sum_{j=1}^{N}w_{ij}x_{ijt}\theta +\mu_{i}+\eta_{t}+\varepsilon_{it}$$
where *ρ* is the spatial autocorrelation coefficient vector of a (*M* × 1) independent variable.

The spatial panel data models described above are generally estimated using the maximum likelihood method [34]. In this study, we first focus on fixed-effect models, including spatial fixed effects, time-period fixed effects, and spatial and time-period fixed effects. We also estimate a random effects model to be thorough, and we utilize the Hausman test to choose the model that fits best.

### 2.3. Input-Output Panel Data

The DMU in this study is cities at the prefecture level in China, so input-output panel data was compiled for 283 such cities between 2003 and 2013. Its specific composition is as follows:

#### 2.3.1. Capital Stock

Capital stock is a widely used input variable [9,12,13,20,22,24,26]. Calculating the capital stock of prefecture-level cities has always been difficult in China, but there has been some progress in recent years. For instance, Ke and Xiang calculated the fixed capital stock of 286 cities at the prefecture level in China between 1996 and 2014 [35]. This data was obtained from the Economic Data Research Center of Hunan University (http://edrc.hnu.edu.cn/).

#### 2.3.2. Labor

Quality of labor and labor time are vital influencing factors of labor input. Due to the difficulty of acquiring data, however, this study uses the total number of employees for prefecture-level cities from the China Regional Economic Statistical Yearbook as the input data for labor. The total number of employees refer to the number of populations who are economically active. This data is also widely accepted for representing the labor force [7,13,24].

#### 2.3.3. Energy

Energy consumption also serves as an important input variable [8,10,13]. However, one difficulty is the lack of unified public data on energy use in China at the prefecture level. Presently, the China Energy Statistical Yearbook only details energy use data at the provincial level and not at the prefecture level. Because of this, Wang and Li [36], as well as Su et al. [37], used DMSP/OLS nighttime light satellite data to describe energy use and emissions from energy consumption in China at the prefecture level between 1992 and 2013, showing that it is possible to incorporate prefecture-level energy consumption into studies on ETFEE. The core idea for energy consumption data is to first obtain spatial data on the built-up area of each prefecture-level city, and use this data to calculate digital number (DN) value information for the DMSP/OLS nighttime light satellite data. This is then combined with existing data on energy consumption of cities to carry out regression analysis on the existing data and total value of the DN to determine the relationship between the total value of the nighttime lights DN and energy consumption. This relationship is used to estimate data for all prefecture-level cities nationwide.

#### 2.3.4. Desirable Outputs

In the majority of studies [9,13,23,26], desirable outputs are expressed in terms of actual GDP. The GDP of each prefecture-level city is unified to calculate the actual GDP value at 2003 constant prices. Original data was drawn from the China Urban Statistical Yearbook.

#### 2.3.5. Undesirable Outputs

Based on existing literature [22,23], this study selected the three indicators of wastewater, sulfur dioxide and soot (dust) as undesirable outputs, with data for the three indicators sourced from the China Urban Statistical Yearbook. As this study adopts the output-oriented DDF model, the weights of the three undesirable output variables depend on the cost of treating pollution [20]. The average cost of treating wastewater is RMB 3/m^3^, while the average cost of treating sulfur dioxide and soot (dust) is RMB 2700/ton (the Beijing Municipal Development and Reform Commission’s Notice on Adjusting the Price of Water for Non-residents in Beijing (2014) stipulated that the sewage treatment fee is 3 yuan per cubic meter. Industrial waste gas treatment cost calculated based on China Environment Statistical Yearbook (2014) data: industrial waste gas treatment operating costs/sulfur dioxide, nitrogen oxide and particulate matter emissions). In 2013, the total volumes of each pollutant were 77.97 million tons of wastewater, 62,588 tons of sulfur dioxide and 31,443 tons of soot (dust). Based on the total treatment costs for each of the indicators, their respective weights were set as 0.4795, 0.3464 and 0.1740.

## 3. Results and Discussion

### 3.1. Energy Total-Factor Efficiency without Environmental Constraints

Under this scenario, the impact of environmental pollution on ETFEE is not considered. Only capital, labor and energy are used as input variables, and GDP is the only output. Overall energy total-factor efficiency between 2003 and 2013 was calculated, and the changes in annual averages are shown in Figure 1. Prefecture-level city averages of energy total-factor efficiency experienced an *N*-shaped trend, with three distinct stages: an upswing from 2003 to 2007, a downswing from 2007 to 2010, and another upswing from 2010 to 2013. This result was confirmed by the research in [22].

Looking at the energy total-factor efficiency of specific prefecture-level cities, Shenzhen, Chengdu, Karamay, Ya’an, Bazhong, Daqing, Jiayuguan, Zunyi, Jinchang and other prefecture-level cities were at the forefront for at least two years (efficiency value = 1). Prefecture-level cities with low efficiency include Xinzhou, Luliang, Yulin, Bozhou, Dingxi, Guyuan, Lincang, Shangluo, Fuyang, Guangyuan, Hanzhong, Pingliang and Weinan. The energy total-factor efficiency of these prefecture-level cities was below 0.2 for at least two years.

The differences between prefecture-level cities are considerable. Luliang has the lowest energy total-factor efficiency value with only about 0.1, while the difference of annual average efficiency value is up to about 0.9. There is no convergence of prefecture-level cities’ ETFEE. Calculations showed that the coefficient of variation of the energy total-factor efficiency of prefecture-level cities gradually increased from 2007 onwards, and that differences between prefecture-level cities have been expanding, with an obvious divergence trend. This result is consistent with the study by Zhang and Choi [8]. Prefecture-level cities with lower energy total-factor efficiency have greater room for improvement.

### 3.2. Ecological Total-Factor Energy Efficiency with Environmental Constraints

Under this scenario, the impact of environmental pollution on ETFEE is considered. Capital, labor, and energy are the input variables, GDP is a desirable output, and wastewater, sulfur dioxide, and soot (dust) are undesirable outputs. Global ETFEE under environmental constraints between 2003 and 2013 has been calculated, and the changes in annual averages are shown in Figure 1. With environmental factors included, there are clear changes in prefecture-level city averages, with four distinct stages: a downswing from 2003 to 2005, an upswing from 2005 to 2008, another downswing from 2008 to 2010, and another upswing from 2010 to 2013.

Looking at the ETFEE of prefecture-level cities, Shenzhen, Xi’an, Karamay, Sanya, Bazhong, Chengdu, Zunyi, Guyuan, Jiayuguan, Jinchang, Wuhai, Ya’an, Haikou, Tongchuan, Anshan, Jiuquan, Ningde and others have been at the forefront for three or more years (efficiency value = 1). Prefecture-level cities with relatively low ETFEE are Baise, Chongzuo, Hechi, Huainan, Laibin, Pu’er, Weinan, Xinzhou, Xinxiang, Chaoyang, Jiaozuo, Lincang and Yuncheng.

Moreover, the results indicate that differences between prefecture-level cities are considerable. The prefecture-level city with the lowest ETFEE was Xinzhou, with only about 0.38, while the difference of annual average efficiency value was up to about 0.62. There was no convergence between the ETFEE of cities. Calculations show that the coefficient of variation of ETFEE between prefecture-level cities displayed a *W*-shaped trend, with the difference between prefecture-level cities gradually shrinking between 2003 and 2006, fluctuating amid stability between 2006 and 2011, and expanding after 2011, with an obvious divergence trend.

Looking at the spatial distribution of ETFEE (see Figure 2), in 2004, prefecture-level cities with high values were mainly concentrated in China’s western region, such as Gansu, Shaanxi, Sichuan, Guizhou and Yunnan provinces; prefecture-level cities with low values were mainly located in northern Hebei, northern Henan, southern Shanxi, central Shaanxi and western Guangxi provinces. By 2007, areas with high values remained largely unchanged, with only the addition of certain parts of Heilongjiang province, improvements in low-value areas of northern Hebei, and some new low-value areas in southwestern Yunnan province. By 2010, the spatial distribution of areas with high values of ETFEE had changed dramatically, with the northeast region hosting the most significant high-value clusters, and parts of Hebei and Shaanxi provinces reverting to low-value areas. By 2013, high-value areas had changed again, with the northeast, Beijing-Tianjin region, Shandong Peninsula, Yangtze River Delta, southern Guangdong province and parts of Inner Mongolia, Gansu and Guizhou becoming high-value areas. Low-value areas, meanwhile, were mainly concentrated in western Guangxi and southwestern Yunnan provinces, though other regions contain scattered low-value areas. The number of prefecture-level cities with low values had decreased, however.

### 3.3. Global Ecological Total-Factor Energy Productivity Index

Table 1 shows changes in the average scores of global ETFEP index and individual indicators with and without environmental constraints. Overall, the mean ETFEP index score without environmental constraints was 1.0154, which indicates a growth rate of 1.54%, and the mean ETFEP index score with environmental constraints was 1.0149, which indicates a growth rate of 1.49%. The fact that the latter rate of increase is less than the former indicates that excessive resource use and environmental pollution hindered the global ETFEP index scores of prefecture-level cities in China, detracting from their ETFEP. However, these growth rates are lower than the other studies, for example, with 2.33% change of GML productivity index in China’s 12 coastal provinces [20]. One possible reason is that different research areas were selected. In fact, the computation result of coastal provinces in my research is also higher than the national average. With or without environmental constraints, technological change (GTCH) is the most important factor in increasing global ETFEP index scores. Mean GPEC and GSEC scores less than 1 indicate that the two factors are obstructive factors. This finding was also confirmed by [20]. This also indicates that the novel methodological system, combined with the GBT, EBM and GML productivity indexes, is suitable and effective.

Moreover, the GTCH value without environmental constraints is greater than with environmental constraints, indicating that technological progress is more difficult under environmental constraints. When environmental constraints are factored in, the GPEC and GSEC scores are significantly higher than the scenario without environmental constraints, indicating that environmental constraints have a significant promotional effect on both. Relevant environmental pollution control systems and management measures have a certain influence. When environmental constraints are factored in, scale efficiency increases, indicating that corresponding economic scale adjustments are more rational.

Looking at inter-annual changes (Figure 3), the ETFEP indexes with and without environmental constraints fell first and then rose between 2004 and 2006, reaching a new high in 2006. The indexes fell sharply between 2006 and 2009, before increasing overall. Without environmental constraints, the GTCH trend was similar to that of ETFEP, while GPEC and GSEC decreased overall before 2010, and then increased overall until 2013. With environmental constraints, GTCH fluctuated, GPEC generally fell prior to 2010 and then generally rose, and GSEC fell overall.

### 3.4. Influencing Factor Regression Results

#### 3.4.1. Model Selection and Diagnosis

Firstly, I utilize the Global Moran’s *I* Index to quantitatively measure the spatial autocorrelation of ETFEE (Figure 4). The test results show that the Moran’s *I* index value was greater than 0 from 2004 to 2013 (including 0.15 in 2004, 0.17 in 2007, 0.20 in 2010 and 0.24 in 2013), and that it was statistically significant (*p* < 0.05), indicating that there was a positive spatial correlation between the high–high cluster and the low–low cluster in the spatial distribution of ETFEE. This result is consistent with other studies [23,24]. It also means that a general econometric model cannot process spatial autocorrelation and spatial dependence, and continuing to utilize the general econometric model results in biased estimates. It is, therefore, necessary to utilize a spatial econometric model that can effectively control spatial autocorrelation and spatial dependence to analyze factors affecting ETFEE and ETFEP.

Before estimating model parameters, it is first necessary to determine which model is the optimal data fitting model. The optimal model was selected in accordance with the model selection steps recommended by LeSage and Pace [38]. Table 2 shows the results of the non-spatial panel data model estimation (the LR-test joint spatial fixed effect and time-period fixed effect were significant, indicating that the model needs to be extended to a two-way fixed effect model. Therefore, we can only report the results of both the spatial and temporal fixed effect models), which are still split into those with and without environmental constraints for comparison. When using the traditional Lagrange multiplier (LM) test, the spatial and time-period fixed effects in the ETFEE model are significant at 1%, except for the spatially autocorrelated error term under environmental constraints, which has no significance, so the null hypothesis of no spatially lagged dependent variable can be rejected. When a robust LM test is used, the null hypothesis of no spatially lagged dependent variable can be rejected at 1%, except for the no spatially autocorrelated error term without environmental constraints, which has no significance. Therefore, in terms of the ETFEE model, a spatial autoregressive regression should be chosen to fit the data, whether with or without environmental constraints. Obviously, the decision as to whether to conduct spatial fixed effects and time-period fixed effects is an extremely vital one.

Moreover, based on the analysis of the likelihood ratio (LR) test for joint spatial and time-period fixed effects, I found that the spatial and time-period fixed effects combined with the non-significant null hypothesis can be rejected, indicating that the model can be extended to have spatial and time-period fixed effects. According to the same model selection method, the spatial error model should be selected whether or not environmental constraints are considered in the ETFEP model, and the model can also be extended to become a two-way fixed effects model.

So far, these tests have indicated that the ETFEE models and the ETFEP models should adopt spatial lag models and spatial error models with two-way fixed effects. Nevertheless, if both the LM test and the robust LM test reject the non-spatial model and choose the spatial lag model or the spatial error model, LeSage and Pace [38] point out that the spatial Durbin model (SDM) should be considered. Based on two null hypothesis tests (generally, the results obtained by estimating the parameters of this model can be used to test the null hypotheses (1) H0: *θ* = 0 and (2) H0: *θ* + *ρβ* = 0), I find that without environmental constraints, in both the ETFEE model and ETFEP model, the null hypothesis that the SDM can be simplified to the spatial lag model can be rejected under the two-way fixed effect. However, it is not possible to reject the null hypothesis that the SDM can be simplified to the spatial error model under the two-way fixed effects. In addition, combined with the LM test and robust LM test results, I found that the spatial lag model better fits the data when environmental constraints are not considered. However, when environmental constraints are considered, both null hypothesis tests can be rejected, so the SDM better fits the data (see Table 3). Moreover, the Hausman test is significant at 1%, indicating that the random effects model should be rejected.

#### 3.4.2. Significant Spatial Spillover of Changes in ETFEE and ETFEP

I now turn to the interpretation of the results on the SAR and SDM with spatial and temporal fixed effects, and I will limit our interpretation to the results of these models. Table 4 shows the final estimated results of the factors affecting ETFEE and ETFEP with and without environmental constraints. For spatial spillover, the spatial lagged coefficient *δ* of the spatial lag model or the spatial autocorrelation coefficient *ρ* of the SDM is significantly positive at 1%, once again illustrating that significant spatial agglomeration and spatial spillover exists in the ETFEE and ETFEP of cities at the prefecture level in China. This finding was also confirmed by the study of Li and Wu [24]. Improvements in the ETFEE and ETFEP of neighboring cities will, therefore, contribute to enhancement in the ETFEE and ETFEP of a particular prefecture-level city. This finding demonstrates that extensive regional cooperation with synergistic improvement in the ETFEE and ETFEP is indispensable. It also implied that the spatial spillover of ETFEE and ETFEP in its current form had been investigated and handled properly.

#### 3.4.3. Analysis of Regression Result for the ETFEE Model

Looking at the estimated results of ETFEE without environmental constraints, energy consumption will not produce undesirable outputs. When a series of urban characteristic variables and fixed effects were controlled, cumulative pure efficiency growth has a significant positive impact on energy efficiency, but cumulative scale efficiency growth has a significant negative effect on ETFEE. Per capita GDP, share of secondary industry, and fiscal decentralization have a significant negative impact on ETFEE, and openness and capital–labor ratio have a significant positive effect on ETFEE; however, cumulative technological progress, fiscal expenditure on scientific undertakings and per capita energy use have no significant effect on ETFEE. This shows that without environment constraints, the main factors contributing to the ETFEE of cities at the prefecture level in China are cumulative pure efficiency growth, capital–labor ratio and FDI.

With environmental constraints, energy consumption produces undesired outputs, and cumulative technological progress, cumulative net efficiency growth and cumulative scale efficiency growth all have a significant positive influence on ETFEE. Per capita GDP, share of secondary industry, per capita energy use and capital–labor ratio, on the other hand, have a significant negative influence on ETFEE. This is related to the stage of China’s industrialization, which is characterized by a high proportion of heavy industry, high levels of energy consumption and considerable investment dependence. These factors are obvious hindrances to improving ETFEE. Fiscal expenditure on scientific undertakings, fiscal decentralization, and FDI do not have a significant influence on ETFEE. This shows that when environmental constraints are considered, the main factors contributing to the ETFEE of cities at the prefecture level in China are cumulative pure efficiency growth, cumulative technological progress and cumulative scale efficiency growth.

#### 3.4.4. Analysis of Regression Result for the ETFEP Model

Looking at the estimated results of ETFEP, with no environmental constraints, cumulative technological progress, cumulative scale efficiency growth, cumulative pure efficiency growth, capital–labor ratio and proportion of secondary industry all have a significant positive influence on the ETFEP index. Meanwhile, per capita GDP and per capita energy use have a significant negative impact on the ETFEP index. Fiscal decentralization and FDI have no significant influence on the ETFEP index. This shows that when environmental constraints are not considered, the main factors contributing to the ETFEP index of cities at the prefecture level in China are cumulative technological progress, cumulative scale efficiency growth, cumulative pure efficiency growth and capital–labor ratio.

When environmental constraints are considered, cumulative technological progress, cumulative scale efficiency growth, cumulative pure efficiency growth, share of secondary industry and capital–labor ratio have a significant positive influence on the ETFEP index, and the other factors have no significant influence on it.

#### 3.4.5. Brief Summary of Regression Results

I can summarize from the above that cumulative technological progress, cumulative scale efficiency growth and cumulative pure efficiency growth rate have a significantly positive influence on ETFEE and the ETFEP index, indicating that these three factors are important sources of improving ETFEE. Per capita GDP has a significantly negative influence on ETFEE and the ETFEP index, indicating that not all economically developed cities have high ETFEE, as these cities often emit large volumes of pollutants to derive their economic gains. Even if environmental constraints are not considered, the ETFEE of these cities is low, so the spatial distribution of economically developed areas does not completely overlap with areas of high ETFEE. Contrary to what one would expect, fiscal expenditure on scientific undertakings has a significant positive influence on increasing ETFEE when environmental constraints are not factored in, but no influence on improving ETFEE in other states, and it does not lead to substantial improvement in ETFEE. This is worth noting. The share of secondary industry has a significant negative influence on improving ETFEE, indicating that increased industrialization is a hindrance to improving ETFEE. This finding was confirmed by the study of Li and Hu [10]. In addition, the necessity of optimizing the secondary industry was also proved. Nevertheless, it has a significant positive impact on ETFEP. Therefore, the relationship between industrialization and ETFEE requires a dialectical understanding [39]. On the one hand, an excessively heavy industrial structure is not conducive to improving ETFEE; on the other hand, higher industrialization encourages overall progress in energy technology. Although FDI is a significant positive factor contributing to the ETFEE in the scenario of without environment constraints, it is not as significant as expected for other three models, which indicates that the technology spillover effect of FDI does not play a role. This finding is not consistent with the province-level study of Li and Hu [10]. Due to differences in data and study scale, they found the FDI is conformed to the “pollution halo” hypothesis. However, this hypothesis is inconclusive for prefecture-level cities. One possible reason for this is that prefecture-level cities located in central and western China have acquired little FDI. Hence, it is implied that this tiny investment has no obvious effects on improving ETFEE and ETFEP. Meanwhile, I also found that the mechanism of fiscal decentralization does not affect ETFEE. A reduction in per capita energy consumption helps to improve ETFEE. The capital–labor ratio has a significant negative effect on improving ETFEE under environmental constraints. However, it has a significant positive impact on improving ETFEE and ETFEP under other scenarios, indicating that the endowment advantages of factors of production generally play a positive role in increasing ETFEE and ETFEP.

#### 3.4.6. The Direct, Indirect and Total Effects of the Influencing Factors

The parameter estimates of the non-spatial model can signify the marginal effect of explanatory variables on dependent variables, but it should be noted that the influence coefficient of the SDM model does not directly mirror the marginal effect of the explanatory variables on the explained variables [34]. Therefore, I report the direct, indirect and total effects of the influencing factors. A direct effect is the influence a certain variable has on the ETFEE of a prefecture-level city. The direct effect of an explanatory variable is different from the estimated coefficient because there is a feedback effect. The feedback effect is generated because its influence on the energy efficiency of an individual prefecture-level city is transmitted to neighboring prefecture-level cities and the influence of neighboring prefecture-level cities is transmitted back to the individual prefecture-level city. Generally, the direct effect coefficient in Table 5 is larger than the parameter estimation value in Table 4, and it is thought that there is a positive feedback effect; otherwise, there is a negative effect. Comparing Table 4 and Table 5, the variables ln*GPEC*, ln*GDPPC*, ln*GDP*2 and ln*energypc* in the ETFEE model all have negative feedback effects, which means that if a prefecture-level city increases its cumulative pure efficiency growth, per capita GDP, share of secondary industry and per capita energy use, it will have a positive short-term effect on increasing the prefecture-level city’s ETFEE. Increasing cumulative technological progress, cumulative pure scale efficiency growth and the capital–labor ratio, will meanwhile have a negative short-term effect on increasing its ETFEE. In the ETFEP model, the variables ln*GTCH*, ln*GPEC*, ln*GSEC*, ln*GDP*2 and ln*percap* all have negative feedback effects, indicating that if a prefecture-level city improves its cumulative technological progress growth level, cumulative pure efficiency growth level, cumulative pure scale growth level, share of secondary industry and capital–labor ratio, it will have a positive short-term effect on increasing the prefecture-level city’s ETFEP.

The influence of neighboring prefecture-level cities is characterized by an indirect effect, namely spatial spillover. The regression result in column two of Table 5 displays that the spillover effect of cumulative technological progress is 0.2344, which indicates that if a prefecture-level city increases its cumulative technological progress rate by 1%, this will result in an increase in ETFEE in neighboring cities of 0.23%; if it increases cumulative pure efficiency growth by 1%, however, it will result in an ETFEE reduction of approximately 0.09% in neighboring prefecture-level cities; if it increases its cumulative pure scale efficiency growth by 1%, it will increase the ETFEE of neighboring prefecture-level cities by approximately 0.13%; if a prefecture-level city increases its per capita GDP and the share of secondary industry and per capita energy use by 1%, it will reduce the ETFEE of neighboring prefecture-level cities by approximately 0.11%, 0.08%, and 0.02%, respectively; and if a prefecture-level city increases its FDI and fiscal decentralization by 1%, it will increase the ETFEE of neighboring prefecture-level cities by approximately 0.01% and 0.06%, respectively. Looking at the model in column four of Table 5, an increase in all the variables will lead to a decrease in the ETFEP of neighboring prefecture-level cities. Cumulative pure efficiency growth and cumulative pure scale efficiency growth are the most prominent examples. If a prefecture-level city increases its cumulative pure efficiency growth or cumulative pure scale efficiency growth by 1%, it will reduce the ETFEP of neighboring prefecture-level cities by approximately 0.11% and 0.09%, respectively.

## 4. Conclusions 

As China’s total carbon emissions have constantly increased and its environmental pollution has worsened in recent years, transforming its pattern of economic growth and achieving high-quality economic development have come to be regarded as core to meeting carbon emissions targets and controlling environmental pollution. To achieve high-quality economic developments, restructure the economy, and win the fight against pollution, the focus must be on promoting high-quality development by increasing ETFEP. Therefore, it is particularly vital to explore and understand temporal and spatial changes in ETFEE and their influencing factors. To this end, this paper constructed an input-output panel dataset for 283 prefecture-level cities in China between 2003 and 2013 to systematically evaluate the ETFEE and ETFEP, as well as their compositions, of those cities, with and without consideration of environmental constraints. Spatial panel data models were also employed to systematically examine factors that influence ETFEE and ETFEP.

The main conclusions from this study are as follows: 

(1) The averages of energy total-factor efficiency of prefecture-level cities have experienced a dynamic change. The differences between prefecture-level cities have been expanding, with an obvious divergence trend. This was mainly due to excessive use of energy and high emissions of environmental pollutants, in particular, significantly hindering the growth of technological progress under environmental constraints. I found that environmental constraints are crucial for evaluation of ETFEE of prefecture-level cities in China. ETFEE and ETFEP had differing performance, and corresponding changing trends, both with and without consideration of environmental constraints. It was also found that ETFEE has a significant spatial agglomeration effect and its spatial pattern has undergone significant changes.

(2) On the whole, the mean ETFEP index score without environmental constraints has a growth rate of 1.54%, and the mean ETFEP index score with environmental constraints has a growth rate of 1.49%. The fact that the latter rate of increase is less than the former indicates that excessive resource use and environmental pollution hindered the global ETFEP index scores of cities at the prefecture level in China, detracting from their ETFEP. I found that GTCH is the most important factor in increasing global ETFEP index scores. Mean GPEC and GSEC scores are less than 1, implying that the two factors are obstructive factors with or without environmental constraints. In addition, technological progress is more difficult under environmental constraints.

(3) In terms of influencing factors, increasing cumulative technological progress, cumulative scale efficiency growth, and cumulative pure efficiency growth are important ways of improving ETFEE and ETFEP. Per capita GDP has a significant negative impact on ETFEE and ETFEP, and the spatial distribution of economically developed areas does not completely overlap with areas of high ecological total-factor energy efficiency. Fiscal expenditure on scientific undertakings does not substantially improve ETFEE. Increasing the level of industrialization hinders improvements to ETFEE. There is no technology spillover effect from FDI, and fiscal decentralization does not affect ETFEE. Reducing per capita energy consumption improves ETFEE. Endowment advantages of factors of production generally play a positive role in improving ETFEE.

(4) In terms of the direct effects of the influencing factors, increasing cumulative pure efficiency growth, per capita GDP, share of secondary industry and per capita energy use have a positive short-term influence on a prefecture-level city’s ETFEE; however, increasing cumulative technological progress, cumulative pure scale efficiency growth and the capital–labor ratio have a negative short-term influence on its ETFEE. In addition, increasing the cumulative technological progress growth level, cumulative pure efficiency growth level, cumulative pure scale growth level, share of secondary industry and capital–labor ratio have a positive short-term effect on its ETFEP.

(5) In terms of spatial spillover effects, cumulative technological progress, cumulative pure scale efficiency growth, FDI and fiscal decentralization have positive spillover effects on improving ETFEE; meanwhile, cumulative pure efficiency growth, per capita GDP, share of secondary industry and per capita energy use have negative spillover effects. Nevertheless, all factors have negative spillover effects on ETFEP, of which, cumulative pure efficiency growth and cumulative pure scale efficiency growth are the most significant.

The above research conclusions have the following important policy implications: 

(1) Improving ETFEP is fundamental to transforming the economic growth pattern of China. Because the result confirmed that the per capita GDP has a significantly negative influence on ETFEE and the ETFEP index. There is an obvious mismatch between ETFEE or ETFEP and economic growth level. China needs to change its economic growth pattern and to pursue high-quality development. This requires abandoning the traditional extensive growth pattern that relies on high investment and high energy use and achieving the transition from relying on inputs of factors of production to improving total-factor productivity.

(2) The regression results demonstrate that the three core influencing factors, technological progress, pure efficiency growth and pure scale efficiency growth, mostly have a significant positive influence on the ETFEE and ETFEP. These results have multiple implications for improving ETFEE and ETFEP. On the one hand, although technological progress plays a major part in improving ETFEE, the role of current technological advances has not been fully utilized. It is, therefore, necessary to continue increasing investment in relevant technological innovations, increasing the proportion of fiscal expenditure spent on science and technology, and increasing government support for low-carbon, energy conservation and emission mitigation technologies. On the other hand, pure efficiency change is an obstacle to raising the ETFEE growth rate, indicating that relevant management and institutional innovations have not played their due role in improving ETFEE. In the future, it will be necessary to strengthen further regional and enterprise management innovations, deepen innovations in management systems, formulate, refine and improve relevant laws, standards and policy systems for preventing and controlling environmental pollution, use environmental regulation measures, utilize fully the environmental pollution taxation system, and establish a regional environmental pollution compensation scheme. The third aspect is to improve scale efficiency, determine optimal economic scales based on environmental constraints and carrying capacities, gradually implement a system of total control of fossil energy consumption, further increase the clean energy utilization rate, and use market-based measures to guide enterprises to determine their optimal production scale so as to reduce redundant production.

(3) For the specific influencing factors of ETFEE and ETFEP change, the corresponding influence mechanism has more specific policy implications. For example, industrialization level characterized by the share of secondary industry shows a significant association with ETFEE and ETFEP. This relation suggests that we need to optimize the industrial structure to control the proportion of heavy industry, draw up a list of approved heavy industries, improve environmental standards for heavy industry, limit the development of heavily polluting and energy-intensive industries, increase the proportion of service sector in the industrial structure, and achieve the economic transition. The technology spillover effect of FDI does not play a role, and the mechanism of fiscal decentralization does not affect ETFEE, which signifies that the changes in ETFEE and ETFEP are not sensitive to FDI and fiscal decentralization policy. However, for per capita energy use, the results indicate that China needs to formulate policies to reduce energy consumption per capita and form a habit of energy saving among its citizens.

(4) Significant agglomeration and spatial spillover exists in the ETFEE and ETFEP, implying that, in the course of improving ETFEE, it is necessary to fully address the role of spatial spillover effects to reduce obstacles to improving ETFEE presented by environmental pollution, devise regional measures to increase ETFEE, and develop mechanisms to create regional synergies and control regional environmental pollution. For example, to prevent and control environmental pollution, it is necessary to develop coordinated joint mechanisms and policies that will maximize benefits to individual cities as well as achieve regional environmental security.

(5) The compound influence of many factors on ETFEE and ETFEP suggests that improving ETFEE is a comprehensive and complex systems process. It is not simply a case of improving the energy system itself; it requires giving adequate consideration to various socio-economic factors, so it is also necessary to construct a comprehensive policy mix. Action in one area alone may not achieve good results, so it is necessary to make comprehensive efforts to utilize beneficial factors and limit and constrain deleterious factors. Prefecture-level cities can also formulate effective and targeted policy measures according to their own situations.

## Figures and Tables

**Figure 1 ijerph-16-03480-f001:**
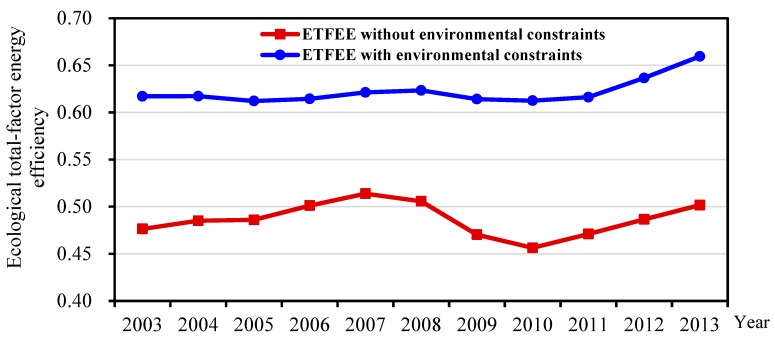
Ecological total-factor energy efficiency variations with and without environmental constraints.

**Figure 2 ijerph-16-03480-f002:**
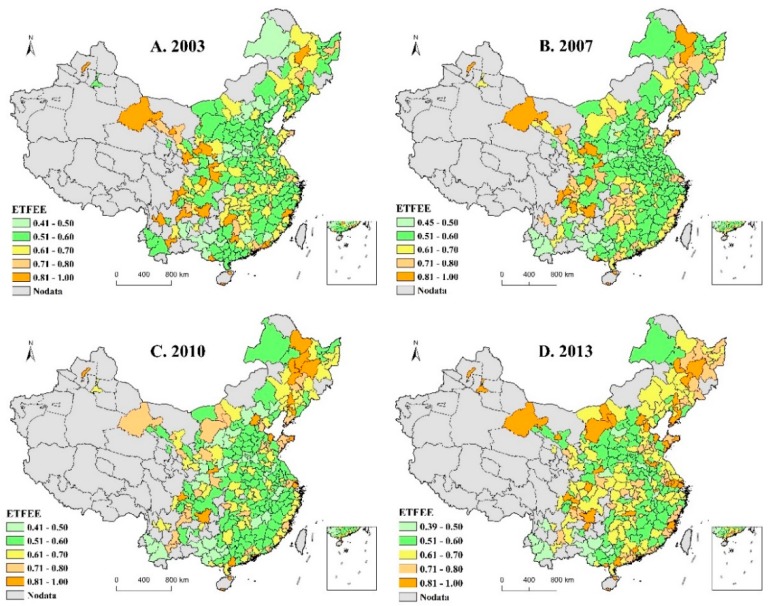
Changes in the spatial distribution pattern of ETFEE.

**Figure 3 ijerph-16-03480-f003:**
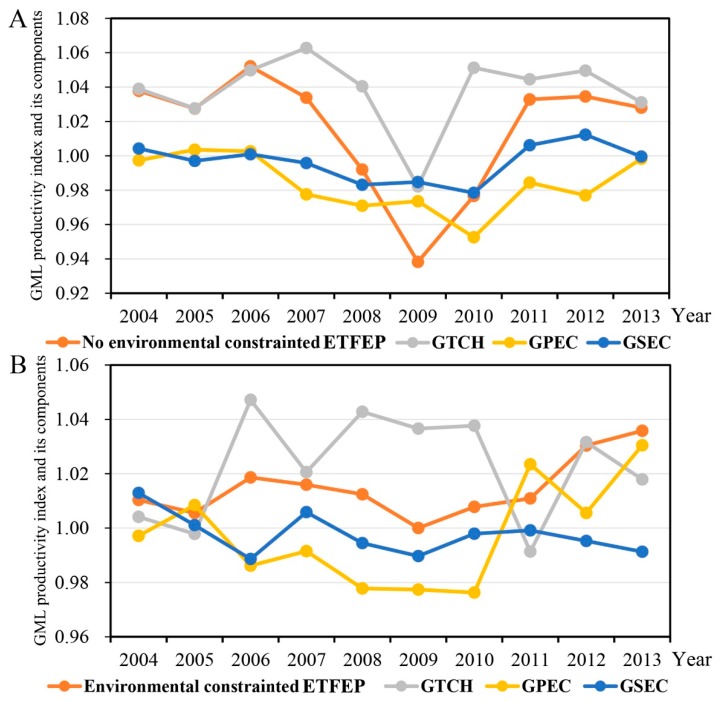
Inter-annual change of ETFEP indexes with and without environmental constraints. (**A**): Changes of ETFEP index and its components with no environmental constrained; (**B**): Changes of ETFEP index and its components with environmental constrained.

**Figure 4 ijerph-16-03480-f004:**
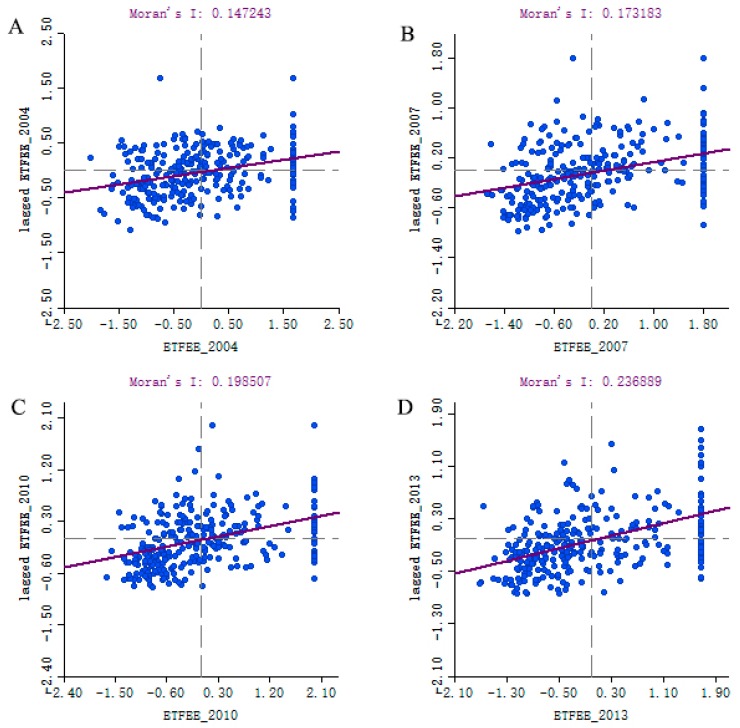
Scatter plots of Moran’s *I* for ETFEE in China’s prefecture-level cities. (**A**): Moran’s *I* for ETFEE in 2004; (**B**): Moran’s *I* for ETFEE in 2007; (**C**): Moran’s *I* for ETFEE in 2010; (**D**) Moran’s *I* for ETFEE in 2013.

**Table 1 ijerph-16-03480-t001:** Comparison of mean values of ecological total factor energy productivity index.

Without Environmental Constraint				With Environmental Constraint			
ETFEP	GTCH	GPEC	GSEC	ETFEP	GTCH	GPEC	GSEC
1.0154	1.0378	0.9838	0.9963	1.0149	1.0226	0.9974	0.9979

Note: ETFEP = ecological total-factor energy productivity; GTCH = technological change; GPEC = pure efficiency change; GSEC = scale efficiency change.

**Table 2 ijerph-16-03480-t002:** Estimation results of non-spatial panel models.

Variables/Parameters	(1)	(2)	(3)	(4)
	Ecological Total-Factor Energy Efficiency Model (ln *ETFEE*)		Ecological Total-Factor Energy Productivity Model (ln *ETFEP*)	
	Without Environmental Constraint	With Environmental Constraint	Without Environmental Constraint	With Environmental Constraint
	Spatial and Time-Period Fixed Effects		Spatial and Time-Period Fixed Effects	
ln*GTCH*	0.3533 ***	0.9524 ***	0.1212 ***	0.3965 ***
ln*GPEC*	0.5548 ***	0.9659 ***	0.1693 ***	0.3499 ***
ln*GSEC*	−0.0002	0.9331 ***	0.2173 ***	0.3535 ***
ln*GDPPC*	0.0997 ***	−0.0243 ***	−0.0285 ***	−0.0153 ***
ln*science*	−0.0132 ***	0.0033 ***	0.0060 ***	0.0030
ln*GDP*2	−0.0883 ***	−0.0693 ***	0.0140	0.0661 ***
ln*FDI*	−0.0065 ***	0.0002	−0.0017	−0.0006
ln*decentralization*	0.0612 ***	0.0131 ***	0.0181 ***	0.0011
ln*energypc*	0.0025	−0.0091 ***	−0.0051 ***	−0.0051 ***
ln*percap*	−0.1359 ***	−0.0184 ***	0.1052 ***	0.0541 ***
*σ* ^2^				
R^2^	0.9252	0.9404	0.4458	0.3054
Log-likelihood	2451	4761	3864	4405
LM spatial lag	38.8413 ***	39.1287 ***	163.2837 ***	37.5431 ***
LM spatial error	33.8966 ***	2.2587	172.6741 ***	75.5010 ***
Robust LM spatial lag	6.0220 ***	87.7167 ***	0.0352	25.3514 ***
Robust LM spatial error	1.0772	50.8467 ***	9.4257 ***	63.3092 ***
LR-test joint spatial fixed effects	5823.0569 ***	6562.8026***	575.1115 ***	375.6227 ***
LR-test joint time-period fixed effects	109.9612 ***	44.0615 ***	558.8533 ***	76.1009 ***

*** Indicates significance at 1% level.

**Table 3 ijerph-16-03480-t003:** Diagnostic tests of spatial specification.

**Parameters**	**Ecological Total-Factor Energy Efficiency Model (ln *ETFEE*)**					
	**Without Environmental Constraint**			**With Environmental Constraint**		
	**Spatial and Time-Period Fixed Effects**	**Spatial and Time-Period Fixed Effects Bias-Corrected**	**Random Spatial Effects, Fixed Time-Period Effects**	**Spatial and Time-Period Fixed Effects**	**Spatial and Time-Period Fixed Effects Bias-Corrected**	**Random Spatial Effects, Fixed Time-Period Effects**
Wald test spatial lag	61.9986 ***	58.2571 ***	57.298 ***	237.5221 ***	213.4409 ***	190.4742 ***
LR test spatial lag	103.3155 ***	103.3155 ***	−1095.8000	231.3311 ***	231.3311 ***	−7289.9000
Wald test spatial error	53.6242 ***	47.799 ***	47.7038 ***	216.666 ***	193.7953 ***	171.0919 ***
LR test spatial error	−45.4531	−45.4531	−29765	74.5876 ***	74.5876 ***	−44515.0000
Hausman test			119.7337 ***			88.3250 ***
**Parameters**	**Ecological Total-Factor Energy Productivity Model (ln *ETFEP*)**					
	**Without Environmental Constraint**			**With Environmental Constraint**		
	**Spatial and Time-Period Fixed Effects**	**Spatial and Time-Period Fixed Effects Bias-Corrected**	**Random Spatial Effects, Fixed Time-Period Effects**	**Spatial and Time-Period Fixed Effects**	**Spatial and Time-Period Fixed Effects Bias-Corrected**	**Random Spatial Effects, Fixed Time-Period Effects**
Wald test spatial lag	103.1596 ***	93.2538 ***	35.9324 ***	73.9652 ***	67.3363 ***	19.1477 **
LR test spatial lag	109.6648 ***	109.6648 ***	4597.9000 ***	78.5143 ***	78.5143 ***	6451.5 ***
Wald test spatial error	89.6711 ***	79.9774 ***	26.4595 ***	67.5344 ***	60.2745 ***	16.6665 *
LR test spatial error	−55.8883	−55.8883	−7655.2000	17.2951 *	17.2951 *	−11478.0000
Hausman test			392.2541 ***			283.3993 ***

* Indicates significance at 10% level. ** Indicates significance at 5% level. *** Indicates significance at 1% level.

**Table 4 ijerph-16-03480-t004:** Estimation results of spatial panel data models.

Variables/Parameters	(1)	(2)	(3)	(4)
	Ecological Total-Factor Energy Efficiency Model (ln *ETFEE*)		Ecological Total-Factor Energy Productivity Model (ln *ETFEP*)	
	Without Environmental Constraint	With Environmental Constraint	Without Environmental Constraint	With Environmental Constraint
ln*GTCH*	0.0030	0.9332 ***	0.2465 ***	0.4107 ***
ln*GPEC*	0.5309 ***	0.9883 ***	0.1410 ***	0.3791 ***
ln*GSEC*	−0.0190 ***	0.9268 ***	0.1866 ***	0.3703 ***
ln*GDPPC*	−0.0093 ***	−0.0209 ***	−0.0323 ***	−0.0087
ln*science*	−0.0009	0.0009	0.0080 ***	0.0033
ln*GDP*2	−0.0104 **	−0.0366 ***	0.0458 ***	0.0872 ***
ln*FDI*	0.0014 ***	0.0007	−0.0011	0.0007
ln*decentralization*	−0.0056 **	−0.0018	0.0001	0.0028
ln*energypc*	0.0004	−0.0055 ***	−0.0113 ***	−0.0024
ln*percap*	0.0163 ***	−0.0168 ***	0.1299 ***	0.0641 ***
W*ln*GTCH*		0.1518 ***		−0.0962
W*ln*GPEC*		−0.1531 ***		−0.1317 ***
W*ln*GSEC*		0.0588		−0.1044 ***
W*ln*GDPPC*		−0.1017 ***		−0.0489 ***
W*ln*science*		0.0001		−0.0020
W*ln*GDP*2		−0.0714 ***		−0.0634 ***
W*ln*FDI*		0.0056 ***		−0.0050 ***
W*ln*decentralization*		0.0542 ***		0.0077
W*ln*energypc*		−0.0145 ***		−0.0079 ***
W*ln*percap*		0.0185		−0.0327 ***
*δ*	0.0232 ***		0.0872 ***	
*ρ*		0.0730 **		0.0612 ***
(Pseudo) Corrected R^2^	0.9088	0.7511	0.1540	0.3299
*σ* ^2^	0.0004	0.0020	0.0039	0.0028
Log-likelihood	7134.7311	4897.8355	3983.8518	4453.6577

* Indicates significance at 10% level. ** Indicates significance at 5% level. *** Indicates significance at 1% level.

**Table 5 ijerph-16-03480-t005:** Direct, indirect and total effects.

Variables/Parameters	(1)	(2)	(3)	(4)
	Energy Efficiency Model (ln *ETFEE*)		Total-Factor Energy Efficiency Model (ln *ETFEP*)	
	Without Environmental Constraint	With Environmental Constraint	Without Environmental Constraint	With Environmental Constraint
**Direct effects**				
ln*GTCH*	0.0031	0.9350 ***	0.2466 ***	0.4093 ***
ln*GPEC*	0.5309 ***	0.9874 ***	0.1417 ***	0.3778 ***
ln*GSEC*	−0.0185 ***	0.9281 ***	0.1862 ***	0.3697 ***
ln*GDPPC*	−0.0093 ***	−0.0221 ***	−0.0328 ***	−0.0094
ln*science*	−0.0009	0.0009	0.0080 ***	0.0033
ln*GDP*2	−0.0103 **	−0.0381 ***	0.0459 ***	0.0862 ***
ln*FDI*	0.0014 ***	0.0007	−0.0011	0.0007
ln*decentralization*	−0.0055 **	−0.0009	0.0001	0.0030
ln*energypc*	0.0004	−0.0056 ***	−0.0113 ***	−0.0025
ln*percap*	0.0163 ***	−0.0165 ***	0.1299 ***	0.0639 ***
**Indirect effects**				
ln*GTCH*	0.0001	0.2344 ***	0.0234 ***	−0.0742
ln*GPEC*	0.0127 **	−0.0858 ***	0.0135 ***	−0.1142 ***
ln*GSEC*	−0.0005	0.1346 ***	0.0177 ***	-0.0872*
ln*GDPPC*	−0.0002	−0.1095 ***	−0.0031**	−0.0523 ***
ln*science*	0.0000	0.0001	0.0008**	−0.0018
ln*GDP*2	−0.0002	−0.0787 ***	0.0044**	−0.0616 ***
ln*FDI*	0.0001 *	0.0061 ***	−0.0001	−0.0052 **
ln*decentralization*	−0.0001	0.0579 ***	0.0000	0.0089
ln*energypc*	0.0000	−0.0159 ***	−0.0011 ***	−0.0085 ***
ln*percap*	0.0004 *	0.0177	0.0123 ***	−0.0309 **
**Total effects**				
ln*GTCH*	0.0032	1.1694 ***	0.2700 ***	0.3352 ***
ln*GPEC*	0.5437 ***	0.9017 ***	0.1552 ***	0.2636 ***
ln*GSEC*	−0.0190 ***	1.0627 ***	0.2039 ***	0.2825 ***
ln*GDPPC*	−0.0095 ***	−0.1316 ***	−0.0360 ***	−0.0617 ***
ln*science*	−0.0009	0.0010	0.0088 ***	0.0015
ln*GDP*2	−0.0105 **	−0.1168 ***	0.0503 ***	0.0246
ln*FDI*	0.0014 ***	0.0068 ***	−0.0012	−0.0046 ***
ln*decentralization*	−0.0057 **	0.0569 ***	0.0000	0.0119
ln*energypc*	0.0004	−0.0216 ***	−0.0124 ***	−0.0110 ***
ln*percap*	0.0167 ***	0.0013	0.1422 ***	0.0330 **

* Indicates significance at 10% level. ** Indicates significance at 5% level. *** Indicates significance at 1% level.

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
