# Peer review of "Spatiotemporal Dynamics of Ecological Total-Factor Energy Efficiency and Their Drivers in China at the Prefecture Level"

_ijerph, 2019, doi:10.3390/ijerph16183480_

Round 1
Reviewer 1 Report
The author combined the epsilon-based measure and Global Malmquist-Luenberger productivity index to evaluate ETFEE and ecological total-factor energy productivity (ETFEP) and its decompositions for 283 prefecture-level cities in China between 2003 and 2013. It is well written and technically sound. I recommend this paper should be published after following comments are addressed.
(1)The definition of “ecological total-factor energy efficiency”, especially “total-factor energy efficiency”, should be clearly stated. Moreover, the definition of “undesirable outputs” and the latest treatment methods in academia should also be clearly stated [1-4].
(2)The author should carefully revise the format of the Reference Part, such as Ref 15, Ref 23, etc. Moreover, the papers published by IJERPH itself should be cited.
(3)The policy implications do not correspond exactly to the factors involved in the previous model. The authors need to carefully modify this section.
[1] Efficiency evaluation of industrial waste gas control in China: A study based on data envelopment analysis (DEA) model. Journal of Cleaner Production 2018, 179, 1–11.
[2] Treating undesirable outputs in DEA: A critical review. Economic Analysis and Policy 2019, 62, 97-104.
[3] Evaluating China’s Air Pollution Control Policy with Extended AQI Indicator System: Example of the Beijing-Tianjin-Hebei Region. Sustainability 2019, 11(3), 939.
[4] Cross-Regional Comparative Study on Environmental–Economic Efficiency and Driving Forces behind Efficiency Improvement in China: A Multistage Perspective. Int. J. Environ. Res. Public Health 2019, 16(7), 1160.
Author Response
Response to Reviewer 1 Comments
Point 1: The definition of “ecological total-factor energy efficiency”, especially “total-factor energy efficiency”, should be clearly stated. Moreover, the definition of “undesirable outputs” and the latest treatment methods in academia should also be clearly stated [1-4].
[1] Efficiency evaluation of industrial waste gas control in China: A study based on data envelopment analysis (DEA) model. Journal of Cleaner Production 2018, 179, 1–11.
[2] Treating undesirable outputs in DEA: A critical review. Economic Analysis and Policy 2019, 62, 97-104.
[3] Evaluating China’s Air Pollution Control Policy with Extended AQI Indicator System: Example of the Beijing-Tianjin-Hebei Region. Sustainability 2019, 11(3), 939.
[4] Cross-Regional Comparative Study on Environmental–Economic Efficiency and Driving Forces behind Efficiency Improvement in China: A Multistage Perspective. Int. J. Environ. Res. Public Health 2019, 16(7), 1160.
Response 1: According to your suggestions, I have added the definitions for “ecological total-factor energy efficiency”, “total-factor energy efficiency”, and “undesirable outputs”. The latest treatment methods in academia was clearly stated.
The ecological total-factor energy efficiency is defined as “the ratio of the target energy input with undesirable outputs to the actual energy input in a region, namely incorporating undesirable outputs into energy efficiency analysis”.
Similarly, the total-factor energy efficiency is defined as “the ratio of the target energy input that is suggested from DEA to the actual energy inputs in a region”.
The undesirable outputs are defined as “it usually relates to externalities, side effects, and adverse impacts produced or resulting from the course of operations, such as emission, pollution, waste, etc.”
The latest treatment methods of undesirable outputs in academia are revised. “There are currently four methods to quantify the influence of environmental pollution emissions (or undesirable outputs) on energy efficiency performance [12-15]. The first is to treat undesirable outputs is to simply disregard them from the production function. The second is to treat environmental pollution as a direct input factor [16]. This method, however, does not reflect real production processes. The third method is to treat undesirable outputs as normal outputs of in the production function. Because this method is more consistent with actual production processes, it has been employed extensively in recent years. The fourth is to transform the undesirable outputs (such as monotone decreasing transformation, data normalization, and linear transformation).”
Point 2: The author should carefully revise the format of the Reference Part, such as Ref 15, Ref 23, etc. Moreover, the papers published by IJERPH itself should be cited.
Response 2: According to your suggestions, I have revised the format of the reference section. Moreover, I also added the references published by IJERPH itself.
Point 3: The policy implications do not correspond exactly to the factors involved in the previous model. The authors need to carefully modify this section.
Response 3: According to your suggestion, I have revised the policy implications to correspond exactly to the factors involved in the previous model. And I have added the related result statements to matching policy implications.
I tried my best to improve the manuscript and made some changes in the manuscript. I appreciate for Editors/Reviewers, warm work earnestly, and hope that the correction will meet with approval.
Once again, thank you very much for your comments and suggestions.
*********************************************************************

Reviewer 2 Report
This paper is not enough clarity about the jurisdiction either you want to target “cities or prefectures”, it is confusing. There are some errors in spelling, and some more clarifications and improvements are needed for reconsidering it for the publication in the International Journal of Environmental Research and Public Health.
In addition to the above, I have a few points for the authors to consider them before the publication of this work:
The abstract is with too much confusion because your results description in abstract composes of long sentences, which is making confusion for the readership of the journal. So, rewrite the abstract with clear and concise sentences without ambiguous illustration. Please highlight your contribution and novelty of this manuscript with accuracy in the introduction part. Please update your literature with few latest studies:
Check whether the following reference is relevant and may be cited:
Rauf, A.; Liu, X.; Amin, W.; Ozturk, I.; Rehman, O.U.; Sarwar, S. Energy and Ecological Sustainability: Challenges and Panoramas in Belt and Road Initiative Countries. Sustainability 2018, 10, 2743.
Chandio, A.A.; Jiang, Y.; Rauf, A.; Mirani, A.A.; Shar, R.U.; Ahmad, F.; Shehzad, K. Does Energy-Growth and Environment Quality Matter for Agriculture Sector in Pakistan or not? An Application of Cointegration Approach. Energies 2019, 12, 1879.
Rauf, A.; Zhang, J.; Li, J.; Amin, W. Structural changes, energy consumption and carbon emissions in China: Empirical evidence from ARDL bound testing model. Struct. Chang. Econ. Dyn. 2018, 47, 194–206.
Please give a description of key research issues condensed into one paragraph highlighting the main points of the research this study deals with. At the end of introduction add a passage which elaborates the structure of current manuscript in section wise. Recheck the references and their style are according to the journal requirements, and in-text and end-text should be same and vice versa. In the discussion section, some more related literature must be added to compare and contrast the key findings with the existing study. The conclusion should be based on your results and discussion. So, do consider it accordingly. The acronyms should be defined at first appearance in the manuscript and then must be consistently used throughout the manuscript. Furthermore, the manuscript must be checked form typo errors and spelling checks.

Author Response
Response to Reviewer 2 Comments
Point 1: This paper is not enough clarity about the jurisdiction either you want to target “cities or prefectures”, it is confusing. There are some errors in spelling, and some more clarifications and improvements are needed for reconsidering it for the publication in the International Journal of Environmental Research and Public Health.
Response 1: According to your suggestions, I added a description jurisdiction for the “cities and prefectures” in China. I also tried my best to handle the problems in spelling errors, ambiguities.
Prefectures formally a kind of prefecture-level divisions as a term in the context of China, are used to refer to an administrative division of China, ranking below a province and above a county in China’s administrative structure. According to the administrative divisions of China, there are three levels of cities, namely provincial-level cities, prefecture-level cities, and county-level cities.
A prefecture-level city is a “city” and “prefecture” that have been merged into one consolidated and unified jurisdiction. As such it is simultaneously a city, which is a municipal entry with subordinate districts, and a prefecture with subordinate county-level cities and counties which is an administrative division of a province. A prefectural level city is often not a “city” in the usual sense of the term (i.e., a large continuous urban settlement), but instead an administrative unit comprising, typically, a main central urban area (a city in the usual sense, usually with the same name as the prefectural level city), and its much larger surrounding rural area containing many smaller cities, towns and villages. In this paper, I used the “prefecture” data to analyse the ecological total-factor energy efficiency of China. In the revised manuscript, I unified the presentation for prefecture-level cities.
Point 2: In addition to the above, I have a few points for the authors to consider them before the publication of this work:
The abstract is with too much confusion because your results description in abstract composes of long sentences, which is making confusion for the readership of the journal. So, rewrite the abstract with clear and concise sentences without ambiguous illustration.
Please highlight your contribution and novelty of this manuscript with accuracy in the introduction part. Please update your literature with few latest studies:
Check whether the following reference is relevant and may be cited:
Rauf, A.; Liu, X.; Amin, W.; Ozturk, I.; Rehman, O.U.; Sarwar, S. Energy and Ecological Sustainability: Challenges and Panoramas in Belt and Road Initiative Countries. Sustainability 2018, 10, 2743.
Chandio, A.A.; Jiang, Y.; Rauf, A.; Mirani, A.A.; Shar, R.U.; Ahmad, F.; Shehzad, K. Does Energy-Growth and Environment Quality Matter for Agriculture Sector in Pakistan or not? An Application of Cointegration Approach. Energies 2019, 12, 1879.
Rauf, A.; Zhang, J.; Li, J.; Amin, W. Structural changes, energy consumption and carbon emissions in China: Empirical evidence from ARDL bound testing model. Struct. Chang. Econ. Dyn. 2018, 47, 194–206.
Response 2: The abstract was rewritten according to your advices. “There is no convergence trend was found in ETFEE between prefecture-level cities. Technical progress plays the largest role in increasing ETFEP growth, pure efficiency change and scale efficiency change are the main hindering factors. Boosting cumulative technological progress, cumulative scale efficiency growth rate and cumulative pure efficiency growth rate are important means of increasing ETFEP. I also found that areas with high levels of economic development do not completely overlap with areas of high ETFEE. Surprisingly, the fiscal expenditure on scientific undertakings and technological spillover effects from FDI have not substantially increased ETFEE. Whereas increased industrialization hinders the improvement of ETFEE. Furthermore, reducing per capita energy consumption help boost ETFEE. And endowment advantages of factors of production have a positive overall effect on improving ETFEE. Lastly, important policy implications are inferred.”
The contribution and novelty of this manuscript was highlighted in the introduction section. I also updated our literature with few latest studies. However, I found the references provided by reviewer is not completely relevant to this paper. Hence, just partly relevant papers were cited.
Point 3: Please give a description of key research issues condensed into one paragraph highlighting the main points of the research this study deals with. At the end of introduction add a passage which elaborates the structure of current manuscript in section wise.
Response 3: I have added one paragraph to describe the key research issues and main points of the research this study deals with. “Based on the above analysis, the key research issues or main points were condensed into several aspects, including characterization of evolutionary process and spatial-temporal pattern of ETFEE and ETFEP in China at the prefecture-level cities, identification and estimation of the influencing factors of these two indices changes, investigation of spatial spillover existed in the variation of ETFEE and ETFEP. Specifically, this study selects labor, capital stock and energy use as input factors [9,13,26], constant-price GDP (gross domestic product) as a desirable output [9,13,26], and wastewater discharge, sulfur dioxide emissions and soot (dust) emissions as undesirable outputs [13,22,23]. Then the EBM and the GML productivity index were applied to calculate ETFEE, ETFEP index and constituent components for 283 prefecture-level cities in China between 2003 and 2013, with consideration given to resource and environmental factors. Finally, I utilize a spatial econometric model (the main purpose is to control the spatial effect to obtain a more reliable regression result) to examine the impact of socio-economic factors on the changes in ETFEE and ETFEP.”
According to your suggestion, a description for structure of this manuscript was added in the end of introduction. “The rest of this paper structured as follows. In Section 2, a detailed description for the materials and methods is presented. Results analysis and related discussion is illustrated in Section 3. Section 4 concludes this paper and provides several important policy implications.”
Point 4: Recheck the references and their style are according to the journal requirements, and in-text and end-text should be same and vice versa.
Response 4: I have rechecked and revised the references and the consistency between in-text and end-text according to the journal requirements.
Point 5: In the discussion section, some more related literature must be added to compare and contrast the key findings with the existing study.
Response 5: I have added the related literatures to compare and contrast the key findings with the existing study.
Point 6: The conclusion should be based on your results and discussion. So, do consider it accordingly.
Response 6: I have revised the conclusion section to correspond to the results and discussion.
Point 7: The acronyms should be defined at first appearance in the manuscript and then must be consistently used throughout the manuscript. Furthermore, the manuscript must be checked form typo errors and spelling checks.
Response 7: According to your suggestions, I have added definition at the first appearance for the acronyms. The form typo errors and spelling checks were addressed.
I tried my best to improve the manuscript and made some changes in the manuscript. I appreciate for Editors/Reviewers, warm work earnestly, and hope that the correction will meet with approval.
Once again, thank you very much for your comments and suggestions.
*********************************************************************

Reviewer 3 Report
Dear Authors
This is an well-written paper dealing with index to evaluate ETFEE and ecological total-factor energy productivity (ETFEP). Also it is remarkable in terms of scope and period from 2003 to 2013.
Although I can’t see any reference issue I would suggest to better specify (line 97) previous literature that lead to factor selection (you correctly did it throughout the paper) “This study focuses on spatial and temporal changes in the ETFEE and ETFEP of 97 prefecture-level 98 cities in China and estimates the main socio-economic factors causing those changes. Specifically, this 99 study selects labor, capital stock and energy use as input factors, constant-price GDP as a desirable 100 output, and wastewater discharge, sulfur dioxide emissions and soot (dust) emissions as undesirable 101 outputs”.
I would suggest you to add some captions and explanations in some graphs e.g. line 330 - please add details (x and y explanations and unit measures)…line 394 etc.
The introductory section is comprehensive; nevertheless, perhaps it contains some information that may be placed in the following section e.g. from line 105 to 133 you properly describe how the paper contributes to the existing literature. I would advise to synthetize such sentences for readiness purposes and eventually better explain in the results and discussion sections.
Please insert some details regarding the data in section 2.3.2. Labor. “297 Quality of labor and labor time are vital influencing factors for labor input. Due to the difficulty 298 of acquiring data, however, this study uses employment data for cities at the prefecture level from 299 the China Regional Economic Statistical Yearbook as the input data for labor”.
Please cite prominent studies you mentioned. “2.3.4. Desirable outputs In the majority of studies, desirable outputs are expressed in terms of actual gross domestic 312 product (GDP). The GDP of each city is unified to calculate the actual GDP value at 2003 constant 313 prices. Original data was drawn from the China Urban Statistical Yearbook”
Results and discussion. I would suggest (but this is up to you) to add a subsection that summarized key findings in order to provide readers with a smoother section to read.
Author Response
Response to Reviewer 3 Comments
Point 1: Although I can’t see any reference issue I would suggest to better specify (line 97) previous literature that lead to factor selection (you correctly did it throughout the paper) “This study focuses on spatial and temporal changes in the ETFEE and ETFEP of 97 prefecture-level 98 cities in China and estimates the main socio-economic factors causing those changes. Specifically, this 99 study selects labor, capital stock and energy use as input factors, constant-price GDP as a desirable 100 output, and wastewater discharge, sulfur dioxide emissions and soot (dust) emissions as undesirable 101 outputs”.
Response 1: According to your suggestion, I have added the specific pervious literatures that lead to factor selection. And this issue was corrected throughout the paper especially in the section of 2.3. Input-output panel data.
Point 2: I would suggest you to add some captions and explanations in some graphs e.g. line 330 - please add details (x and y explanations and unit measures)…line 394 etc.
Response 2: According to your suggestion, the captions and explanations was added in the revised figures.
Point 3: The introductory section is comprehensive; nevertheless, perhaps it contains some information that may be placed in the following section e.g. from line 105 to 133 you properly describe how the paper contributes to the existing literature. I would advise to synthetize such sentences for readiness purposes and eventually better explain in the results and discussion sections.
Response 3: According to your suggestion, I have rewritten the introduction section especially the shortcomings of previous studies and contributes to the existing literature. A description of limitations in previous studies was added. Then to cope with the above limitations, the contributions and novelties of this research were proposed. Therewith, the key research issues or main points were condensed. I think current structure arrangement and innovative description fits the general writing habits and reader’s reading habits. And a clear statement of innovations and contributions in introduction section often is the basic requirement of journals and reviewers based on my experiences, previous papers and literatures. Hence, I did not change these sentences to the results and discussion section. But in the results and discussion section, I have highlighted these sentences for readiness purposes and eventually better explain. Moreover, in order to avoid confusion, line 88-96 was changed to materials and methods section. Thanks again for your comments.
Point 4: Please insert some details regarding the data in section 2.3.2. Labor. “297 Quality of labor and labor time are vital influencing factors for labor input. Due to the difficulty 298 of acquiring data, however, this study uses employment data for cities at the prefecture level from 299 the China Regional Economic Statistical Yearbook as the input data for labor”.
Response 4: I have added a detailed describe for labor data used in this paper. “The total number of employees refer to the number of populations who are economically active. This data also is widely accepted data to represent labor force [5,11,28].”
Point 5: Please cite prominent studies you mentioned. “2.3.4. Desirable outputs In the majority of studies, desirable outputs are expressed in terms of actual gross domestic 312 product (GDP). The GDP of each city is unified to calculate the actual GDP value at 2003 constant 313 prices. Original data was drawn from the China Urban Statistical Yearbook”
Response 5: According to your suggestions, I have cited four latest literatures [7,11,30,35].
Point 6: Results and discussion. I would suggest (but this is up to you) to add a subsection that summarized key findings in order to provide readers with a smoother section to read.
Response 6: According to your suggestions, I have added six subsubsection to provide readers with a smoother section to read (please see the section 3.4).
I tried my best to improve the manuscript and made some changes in the manuscript. I appreciate for Editors/Reviewers, warm work earnestly, and hope that the correction will meet with approval.
Once again, thank you very much for your comments and suggestions.
*********************************************************************

Round 2
Reviewer 1 Report
The author has addressed the comments in the revised version of manuscript. The current version is good to be published.
Reviewer 2 Report
Paper is hugely improved and expressed as a case in an appropriate way. The current paper has an acceptable language style for the readership of IJERPH. Hence, I accepted this paper for the final publication. Research implications are properly written and, so far this paper is accepted in recent format for final publication in IJERPH.
Hence, the current version of the manuscript is well documented and adequately justify the publication of this paper in IJERPH.
Recommendation: Accept the manuscript in the current format.